# SGDC: Structurally-Guided Dynamic Convolution for Medical Image Segmentation

**Bo Shi** [iD]                                    BO.SHI@MAIL.CONCORDIA.CA
**Wei-ping Zhu** [iD]                           WEIPING@ECE.CONCORDIA.CA
**M.N.S Swamy** [iD]                          SWAMY@ENCS.CONCORDIA.CA
*Department of Electrical and Computer Engineering, Concordia University*

**Editors:** Accepted for publication at MIDL 2026

## Abstract

Spatially variant dynamic convolution provides a principled approach of integrating spatial adaptivity into deep neural networks. However, mainstream designs in medical segmentation commonly generate dynamic kernels through average pooling, which implicitly collapses high-frequency spatial details into a coarse, spatially-compressed representation, leading to over-smoothed predictions that degrade the fidelity of fine-grained clinical structures. To address this limitation, we propose a novel Structure-Guided Dynamic Convolution (SGDC) mechanism, which leverages an explicitly supervised structure-extraction branch to guide the generation of dynamic kernels and gating signals for structure-aware feature modulation. Specifically, the high-fidelity boundary information from this auxiliary branch is fused with semantic features to enable spatially-precise feature modulation. By replacing context aggregation with pixel-wise structural guidance, the proposed design effectively prevents the information loss introduced by average pooling. Experimental results show that SGDC achieves state-of-the-art performance on ISIC 2016, PH2, ISIC 2018, and CoNIC datasets, delivering superior boundary fidelity by reducing the Hausdorff Distance (HD95) by 2.05, and providing consistent IoU gains of 0.99%-1.49% over pooling-based baselines. Moreover, the mechanism exhibits strong potential for extension to other fine-grained, structure-sensitive vision tasks, such as small-object detection, offering a principled solution for preserving structural integrity in medical image analysis. To facilitate reproducibility and encourage further research, the implementation code for both our SGE and SGDC modules has been is publicly released at https://github.com/solstice0621/SGDC.

**Keywords:** Medical Image Segmentation, Dynamic Convolution, Structure Preservation, Feature Refinement, Boundary Awareness

## 1. Introduction

A core challenge in medical image segmentation is the trade-off between expanding the network's receptive field (Chen et al., 2017) for semantic understanding and preserving high spatial resolution for precise boundary delineation. Operations designed for the former often compromise the latter, leading to fidelity loss of intricate structures critical for clinical diagnosis.

One prominent line of research introduces explicit edge supervision (Lin et al., 2023, 2025), typically employing an auxiliary branch to learn boundary information for guiding the main segmentation network. While widely adopted, the utility of this guidance is fundamentally constrained by the fusion mechanisms employed. Strategies that compress

boundary features into a single-channel attention map impose an inherent information bottleneck, preventing the transmission of rich structural cues. Conversely, methods relying on simple concatenation or addition risk having these high-frequency details smoothed out by subsequent convolutional operations. Therefore, preserving structural fidelity requires a deeper and more interactive fusion mechanism capable of effectively modulating semantic features with structural priors, rather than simple aggregation.

A more advanced approach is spatially variant dynamic convolution (Chen et al., 2020; Lou and Yu, 2025), which employs a kernel prediction network to generate unique convolutional parameters for each spatial location. This enables highly adaptive, location-aware feature extraction and enhances local structural modeling. Existing methods in this paradigm focus on semantic awareness, generating guidance either from the local feature neighborhood itself (Li et al., 2021) or from features at different network stages (Lou and Yu, 2025). Despite their sophistication, these methods share a critical flaw: reliance on adaptive average pooling to generate dynamic kernels. This operation discards pixel-level positional information essential for boundary handling, causing the generated kernels to depend on coarse spatially ambiguous signal, which ultimately leads to over-smoothing of fine details.

Beyond the pooling operation, a more fundamental limitation lies in the intrinsic nature of the conditioning signal itself. In existing paradigms, the guidance is implicitly derived from the network's own backbone features (Wang et al., 2025). These features are inherently trained to maximize intra-class consistency for semantic recognition, resulting in spatially smooth representations dominated by low-frequency information. Using such semantically homogeneous signals to generate dynamic kernels creates a paradox: the kernels are tasked with refining boundaries, yet they are conditioned on features that have already suppressed structural details. Consequently, the generated kernels remain insensitive to fine geometric variations, such as thin boundaries or irregular contours, regardless of the spatial resolution.

These limitations indicate that semantic features alone are insufficient for driving spatially adaptive kernels, especially when boundary precision is required. Instead of relying on implicit conditioning signals (Li et al., 2021) derived from pooled or fused semantics, the dynamic mechanism benefits from a separate source that explicitly encodes structural cues. Incorporating a supervised boundary-guidance stream provides high-frequency information that the semantic pathway cannot supply, allowing the kernel generation process to respond to local geometric variations that are essential for accurate medical segmentation.

We propose a novel structure-guided dynamic convolution (SGDC) network (SGD-Net), a framework designed to directly address this structural information loss through two core innovations. First, we introduce an average-pooling-free SGDC mechanism, which leverages explicit, high-fidelity structural guidance, rather than pooled semantic information, to generate spatially-aware dynamic kernels and gating signals, thereby enabling adaptive feature refinement sensitive to boundary details. Second, we propose a structure guidance extractor (SGE), an independently supervised auxiliary branch that produces multi-channel guidance maps from multi-scale features, delivering the high-frequency structural cues needed by our SGDC modules.

## 2. Related Works

### 2.1. Dynamic Convolution

Dynamic convolution has evolved significantly as a powerful mechanism to enhance the representational capacity of neural networks by adapting model parameters to the input. The early stage of this evolution, characterized by sample-variant methods such as Cond-Conv (Yang et al., 2019) and DynamicConv (Chen et al., 2020), introduced the concept of generating a unique set of kernels for each input image. However, these methods primarily rely on Global Average Pooling (GAP) to aggregate a global context vector for kernel generation. While effective for image classification, this design is inherently spatially invariant; it applies the same dynamic kernel across the entire image, treating background and fine-grained foreground structures identically, which is suboptimal for dense prediction tasks like segmentation that require spatial precision.

To address the localization bottlenecks of sample-variant methods, the paradigm has shifted towards spatially variant dynamic convolutions, which can be categorized into two streams based on their guidance source. The first stream, represented by methods, such as Involution (Li et al., 2021), Condconv(Yang et al., 2019) and (Wang et al., 2025) utilizes implicit self-guidance, where kernel generation is conditioned solely on the local neighborhood of the target pixel, or even derived strictly from the individual pixel instance to avoid spatial aggregation. The second stream employs implicit heterogeneous-source guidance, implemented via the Contmix dynamic convolution module (Lou and Yu, 2025) in the OverLock network. Contmix constructs conditioning signals by aggregating feature combinations across different hierarchical levels of the network. However, despite its improved spatial adaptivity, this design shares a critical limitation when applied to medical segmentation: the prevalent reliance on adaptive average pooling for context compression, which implicitly collapses high-frequency spatial details into a coarse representation. The conditioning signals are derived from the network's semantic backbone, optimized for intra-class consistency. This inherent homogeneity suppresses the high-frequency variations necessary for precise boundary delineation. Consequently, driving dynamic kernels with these semantically pooled signals creates a fundamental paradox: the refinement mechanism is effectively blind to the fine structures it is intended to preserve.

### 2.2. Explicit Structural Guidance in Segmentation

Parallel to dynamic convolution, explicit edge supervision methods (Lin et al., 2023, 2025) adopt a multi-task framework to learn boundary information explicitly, effectively providing valuable high-frequency cues for structural fidelity. However, their effectiveness is fundamentally constrained by the inefficient fusion mechanism. Common strategies—such as compressing boundary features into a single-channel attention map—create a severe information bottleneck. Furthermore, simple element-wise addition or concatenation often fails to effectively modulate semantic features, leading to the washing out of sharp edge details by subsequent layers. Therefore, despite successfully extracting edges, these approaches lack a potent mechanism to fully leverage this structural information for effective feature refinement.

## 3. Methodology

### 3.1. Overall Architecture

Our proposed SGD-Net, illustrated in Fig. 1 (a), is built upon a hierarchical encoder–decoder architecture. To achieve robust feature extraction, we employ a Res2Net-50 backbone pre-trained on ImageNet to generate multi-scale feature maps. The shallowest features are further processed by a Transformer encoder variant (Lin et al., 2025) to capture long-range dependencies. This encoder employs a sparse sampling strategy with multi-scale strides (16, 16), (8, 8), (4, 4), (2, 2), which dramatically reduces sequence length, contributing only 8.07 GFLOPs to the model complexity. Similarly, a Dual Attention module is applied solely at the deepest encoder layer (8x8 resolution) to enhance global semantic context, adding a mere 2.01 GFLOPs. The core innovations of our network involve the SGE and SGDC modules, shown in Fig. 1 (b) and (c), respectively, which are utilized to refine these multi-scale features before passing them to a decoder specifically designed for boundary recovery. Instead of simple concatenation, we employ a reverse attention mechanism. At each decoding stage, this module utilizes the upsampled high-level semantic features, denoted as $\mathbf{F}_{up}$, to generate a reverse attention mask formulated as $1 - \sigma(\mathbf{F}_{up})$. This mask is applied to the shallow features to suppress the most salient regions (typically the lesion center) and force the network to focus on mining fine-grained structural details at the boundaries.

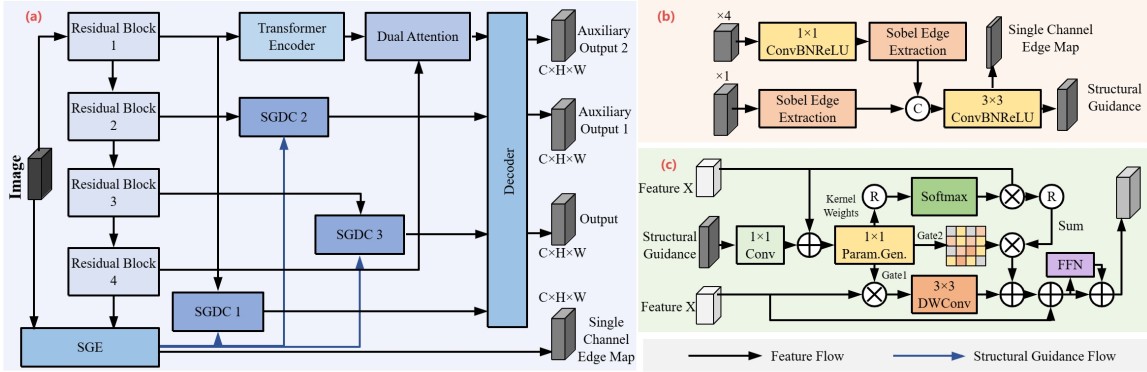

Figure 1: (a) The illustration of our proposed SGD-Net. (b) Structures of SGE Module. (c) structures of SGDC Module

To ensure a comprehensive and effective training, our network is optimized via a multi-output supervision strategy. The decoder produces segmentation predictions at three distinct scales. Outputs from coarser scales, referred to as auxiliary outputs, provide intermediate supervision signals. This deep supervision technique ensures that feature representations across all network depths are effectively optimized.

Furthermore, our SGE generates a dedicated edge map. This output is directly supervised by ground-truth boundary maps, compelling the SGE to explicitly learn high-fidelity structural guidance. This ensures that the guidance provided to our core module SGDC is both accurate and rich in essential boundary details.

### 3.2. Structure Guidance Extractor

The SGE module, detailed in Fig. 1 (b), is designed to generate the high-fidelity structural guidance essential for our SGDC mechanism. Its core innovation lies in the decoupling of two distinct tasks: generating a single channel edge map for explicit supervision and a multi-channel structural guidance feature for the SGDC modules. This design is crucial because the ideal representations for these tasks differ; supervision requires sharp, precise contours, whereas guidance benefits from richer, multi-dimensional information. The SGE workflow begins by processing features from two distinct network stages: the deep, semantic-rich features from the final encoder block and the shallow, high-resolution features from the first encoder. To align their channel dimensions and ensure computational efficiency, the deep features are first projected into a lower-dimensional space via a $1 \times 1$ convolution. Both feature streams are then processed to extract structural details. In contrast to methods using learnable convolutions for this task, we employ a classic, non-trainable Sobel operator whose fixed kernels explicitly inject a stable and efficient structural guidance into the network, obviating the need to learn basic gradient extraction from scratch. We consciously employ a deterministic Sobel operator over learnable convolutions for this stage. While learnable layers offer flexibility, they are prone to overfitting to specific semantic textures in deep networks, potentially diluting the purity of boundary cues. The fixed Sobel operator acts as a 'structural anchor', ensuring that the extracted guidance remains strictly faithful to geometric gradients regardless of the varying semantic contexts. This injects a stable, domain-independent structural prior into the dynamic mechanism. This process is enhanced through an "Edge Modulation" operation, formulated as:

$$F_{mod} = F_{in} \odot \sigma \left( \sqrt{(F_{in} * K_x)^2 + (F_{in} * K_y)^2} \right) \tag{1}$$

where $F_{in}$ is the input feature, $*$ denotes convolution with the Sobel kernels $(K_x, K_y)$, $\odot$ the element-wise multiplication, and $\sigma$ is the Sigmoid function. This operation selectively amplifies feature activations corresponding to edges. During the fusion stage, the processed deep and shallow features are concatenated. This combined feature map is then passed through a $3 \times 3$ convolution to promote effective interaction and integration. Finally, this refined feature is fed into two independent convolutional heads to generate the decoupled outputs. The single channel edge map is directly supervised by ground-truth boundary maps to ensure localization accuracy, while the Structural Guidance is passed to the SGDC modules, providing the essential high-frequency information that enables effective boundary-aware feature refinement.

### 3.3. Structure-Guided Dynamic Convolution Module

Standard dynamic convolutions, despite their spatial adaptability, are fundamentally limited by their reliance on pooling operations that degrade structural details. To overcome this, we propose the SGDC, a novel feature refinement unit that replaces the conventional pooling-driven method with a guidance-driven one. As illustrated in Fig. 1 (c), the SGDC is a feature processing block centered around a lightweight parameterization head that generates adaptive weights for a parallel dual-branch execution body.

The SGDC module takes two inputs: the main feature map from the encoder, Feature X ($F_X$), and the multi-channel structural guidance ($F_{guidance}$) from the SGE. These features are fused via element-wise addition to form the input to the parameter generation head, a single 1x1 convolution. It dynamically generates three outputs for each spatial location: the dynamic kernel weights ($W_{dyn}$), and two gating signals, Gate1 ($g_1$) and Gate2 ($g_2$).

This dual-branch body processes the input feature $F_X$ through two complementary parallel branches. Before branching, it is crucial to normalize $F_X$ by a LayerNorm layer. The main dynamic branch applies an 'unfold' operation to decompose the normalized feature map into overlapping local patches. The generated kernel weights ($W_{dyn}$) are passed through a 'Softmax' function to act as spatially-varying weighting coefficients. A weighted sum of the patches is then computed—a process equivalent to a large-kernel, spatially variant convolution—and the result is recomposed via a 'fold' operation and modulated by Gate2 ($g_2$). The second branch, a local refinement path, modulates the normalized feature with Gate1 ($g_1$) and processes it with a 3x3 depthwise convolution.

The outputs of the two branches are then merged by element-wise summation, and this branch-level fusion is added back to the original $F_X$ through the primary residual connection. This dual-branch design is motivated by the principle of complementary feature processing. The dynamic branch, driven by the SGE, excels at content-adaptive modulation and long-range structural modeling. However, purely dynamic operators can sometimes introduce spatial instability or over-react in homogenous regions. The local refinement branch, employing a static depthwise convolution, provides a deterministic high-frequency pass. By aggregating these two streams, the SGDC ensures that the feature map benefits from adaptive structural shaping while retaining the fundamental textural integrity and training stability provided by standard convolution. The feature map passes through a second LayerNorm layer and into a Feed-Forward Network (FFN), which consists of two 1x1 convolutions with an intermediate GELU activation. A final residual connection integrates the FFN output. The entire operation can be concisely formulated as:

$$F_X' = \text{LayerNorm}(F_X) \tag{2}$$

$$F_{mix} = \text{dynamic}(F_X', W_{dyn}, g_2) + \text{local}(F_X', g_1) \tag{3}$$

$$F_{out} = \text{FFN}(\text{LayerNorm}(F_X + F_{mix})) + (F_X + F_{mix}) \tag{4}$$

The SGDC module exhibits functional complementarity across scales. At fine resolutions, the guidance signal primarily drives detail refinement and edge sharpening. Conversely, at coarse semantic levels, it prevents the over-smoothing typical of pooling operations, thereby maintaining topological integrity and separating distinct semantic regions even in deep layers.

### 3.4. Loss Function

To ensure comprehensive supervision for both semantic representation and structural fidelity, we optimize SGD-Net end-to-end using a composite multi-task objective function. This objective integrates deep supervision on the multi-scale segmentation predictions with

explicit boundary supervision on the structural priors. The total loss $\mathcal{L}_{\text{total}}$ is formulated as:

$$\mathcal{L}_{\text{total}} = \sum_{i=1}^{3} \mathcal{L}_{\text{seg}}(o_i, M_{\text{gt}}) + \lambda \cdot \mathcal{L}_{\text{edge}}(o_e, E_{\text{gt}}) \tag{5}$$

where the first term represents the segmentation loss. To facilitate robust gradient flow and accelerate convergence, we employ a deep supervision strategy, applying the loss to predictions $\{o_1, o_2, o_3\}$ from three different decoder scales against the ground truth mask $M_{\text{gt}}$. The term $\mathcal{L}_{\text{seg}}$ itself is a hybrid loss combining Binary Cross-Entropy (BCE) and Dice loss, designed to handle pixel-level classification accuracy while mitigating class imbalance issues inherent in medical segmentation.

The second term, $\mathcal{L}_{\text{edge}}$, provides direct supervision to the single-channel edge map $o_e$ generated by the SGE module. We employ a Dice loss for this term to enforce high-fidelity alignment between the predicted structural priors and the ground truth boundaries $E_{\text{gt}}$. The hyperparameter $\lambda$ acts as a trade-off factor to balance the semantic and structural learning objectives. In our experiments, we empirically set $\lambda = 3$ to emphasize the importance of boundary preservation.

## 4. Experiments and Results

### 4.1. Implementation Details

We evaluate SGD-Net on several public datasets: ISIC 2016 (Gutman et al., 2016), ISIC 2018 (Codella et al., 2019), and PH2 (Mendonça et al., 2013) for skin lesion segmentation, and the CoNIC dataset (Graham et al., 2021b) for nuclei segmentation. For ISIC 2018, we report 5-fold cross-validation results. For ISIC 2016 and PH2, to evaluate cross-dataset generalization, the model is trained on ISIC 2016 and tested on PH2. The CoNIC dataset is split into training, validation, and test sets with a ratio of 7:1:2. We use the Dice Coefficient and Intersection over Union (IoU) as primary evaluation metrics. Model efficiency is assessed using FLOPs and the number of parameters.

The model is optimized using the Adam optimizer with an initial learning rate of $1 \times 10^{-4}$ and a polynomial learning rate decay. We use a batch size of 16 and the input image size is $256 \times 256$. The encoder is initialized with ImageNet-pre-trained Res2Net-50 (Gao et al., 2019) weights and fine-tuned for 50 epochs. To facilitate reproducibility, we specify key settings: the boundary ground truth $E_{gt}$ is generated via morphological gradient operations(dilation minus erosion), yielding a fixed 3-pixel thickness. Inside the SGDC module, the dynamic unfold-fold operation utilizes a $7 \times 7$ neighborhood kernel. Data augmentation follows standard protocols (random flips and $\pm 20°$ rotations) using Albumentations. All experiments were conducted on a single NVIDIA L4 GPU. To ensure a fair comparison, we adopt the experimental setup from (Lin et al., 2023) as our primary baseline.

### 4.2. Experiment Results

**Comparisons with State-of-the-Art Methods:** As shown in Table 1 and Table 2, SGD-Net achieves state-of-the-art performance on skin lesion segmentation. On the ISIC

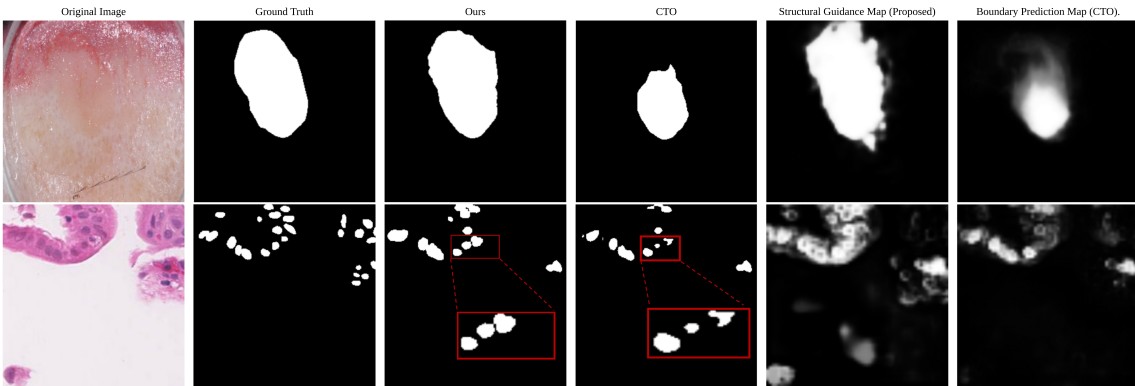

Figure 2: Comparison of qualitative results and structural guidance maps on ISIC 2018(Codella et al., 2019) and CoNIC(Graham et al., 2021b)

Table 1: Dice and IoU comparisons with other methods on ISIC 2016(Gutman et al., 2016) and PH2 (Mendonça et al., 2013) dataset.

| ISIC 2016 & PH2 | | |
|---|---|---|
| **Methods** | **Dice ↑** | **IoU ↑** |
| MSCA (Bi et al., 2016) | 81.57 | 72.33 |
| Bi *et al.* (Bi et al., 2017) | 90.66 | 83.99 |
| Lee *et al.* (Lee et al., 2020) | 91.84 | 84.30 |
| CTO(Lin et al., 2025) | 92.56 | 86.60 |
| **SGD-Net (Proposed)** | **92.93** | **87.29** |

2016→PH2 test, it attains 92.93% Dice and 87.29% IoU, outperforming the best previous method by 0.37% and 0.69%, respectively. On ISIC 2018, our method achieves 91.41% Dice and 84.96% IoU, consistently outperforming classical architectures like TransUNet and CTO. Notably, SGD-Net also surpasses recent competitive models, including UNet v2 (90.47% Dice) and Mamba-based approach VM-UNet V2 (90.79% Dice). Furthermore, the standard deviation from our 5-fold cross-validation indicates the robustness of our method. Despite having fewer parameters than large-scale models like TransUNet (75M vs. 114M), SGD-Net achieves a significant improvements in dice and mean IoU. This confirms that the performance gain stems from the proposed SGDC mechanism rather than mere model capacity scaling.

On the CoNIC challenge (Table 3), in comparison with other methods, SGD-Net achieves 81.61% Dice, 69.46% IoU, and 68.79% PQ, ranking first across all three metrics. In terms of model efficiency, our network achieves these improvements with comparable FLOPs and number of parameters.

Table 2: Dice and IoU comparisons with other state-of-the-art methods on ISIC 2018 dataset (Codella et al., 2019), along with model complexity metrics.

| Methods | Params (M) | GFLOPs | Dice ↑ | IoU ↑ |
|---|---|---|---|---|
| U-Net(Ronneberger et al., 2015) | 7.78 | 14.59 | 82.92 | 77.56 |
| LeViT(Graham et al., 2021a) | 64.23 | 23.57 | 86.98 | 79.68 |
| TransUNet (Chen et al., 2024) | 114.65 | 37.02 | 89.97 | 83.32 |
| UNet v2(Peng et al., 2025) | 25.15 | 5.40 | 90.47 | 82.63 |
| CTO(Lin et al., 2025) | 62.22 | 22.70 | 90.63 | 83.97 |
| VM-UNet V2(Zhang et al., 2024) | 17.91 | 4.40 | 90.79 | 83.31 |
| **SGD-Net (Proposed)** | 75.88 | 36.23 | **91.41 ± 0.24** | **84.96 ± 0.13** |

Table 3: Performance comparison on CoNIC dataset (Graham et al., 2021b).

| Methods | Dice ↑ | IoU ↑ | PQ ↑ |
|---|---|---|---|
| U-Net (Ronneberger et al., 2015) | 78.36 | 66.39 | 64.41 |
| Att-UNet (Schlemper et al., 2019) | 79.44 | 66.11 | 65.28 |
| TransUNet (Chen et al., 2024) | 77.03 | 64.35 | 64.02 |
| CTO (Lin et al., 2025) | 80.67 | 67.97 | 67.17 |
| SGD-Net (Proposed) | **81.61** | **69.46** | **68.79** |

Figure 2 provides a qualitative comparison on the ISIC 2018 and CoNIC datasets, highlighting the superior performance and interpretability of our method. In the top row, where the skin lesion exhibits extremely low contrast against the surrounding tissue, the baseline (CTO) fails to discern the weak transition, resulting in significant under-segmentation. In contrast, SGD-Net accurately delineates the complete lesion structure. Similarly, for the dense nuclei in the bottom row, as highlighted in the red zoomed-in boxes, our method preserves fine-grained geometric topology, whereas the baseline produces blurred and fragmented masks. This performance gap can be explained using the intermediate maps: our Structural Guidance Map (Col. 5) generates sharp, high-contrast responses that precisely highlight structural gradients, whereas the baseline's boundary prediction (Col. 6) appears spatially diffuse and coarse. This visual evidence confirms that our SGE successfully extracts high-fidelity structural priors, which the SGDC module then leverages to prevent the over-smoothing characteristic of pooling-based paradigms. The observed improvements are primarily driven by precise boundary delineation and the effective suppression of false positives in complex backgrounds. These gains stem from the synergy between the SGE, SGDC, and the Decoder. It is also worth-mentioning that the SGE employs semantic modulation to ensure that structural extraction is activated only when supported by deep semantic features, thereby filtering out non-semantic gradients (e.g., hair or texture) before they enter the dynamic mechanism. This guidance subsequently drives the SGDC to modulate features based on valid structural cues, in conjunction with the Decoder's reverse

Table 4: Ablation study of different SGD-Net variants on ISIC 2018 (Codella et al., 2019).

| Method | Dice ↑ | IoU ↑ | HD95 ↓ |
|---|---|---|---|
| (1) Res2Net | 89.13 | 82.42 | 30.54 |
| (2) Baseline (Res2Net + ViT) | 89.57 | 82.91 | 32.63 |
| (3) Baseline+ SGE | 90.39 | 83.43 | 18.58 |
| (4) Baseline+ SGE+ Contmix | 90.95 | 84.22 | 24.21 |
| (5) SGD-Net (Proposed) | **91.41** | **84.96** | **16.09** |

Table 5: Internal mechanism ablation of the SGDC module on ISIC 2018. We analyze the impact of the guidance source and the dual-branch architecture. 'Self-Guidance' implies using the input feature itself for attention generation, simulating standard dynamic convolutions.

| Method | Dice ↑ | IoU ↑ | HD95 ↓ |
|---|---|---|---|
| SGD-Net (Ours) | **91.41** | **84.96** | **16.09** |
| w/o Boundary Supervision | 90.37 | 83.25 | 23.51 |
| w/ Self-Guidance ($F_X$) | 90.52 | 83.65 | 21.34 |
| w/o Local Branch | 90.88 | 84.10 | 18.95 |
| w/o Dynamic Branch | 89.92 | 82.55 | 25.12 |

attention mechanism. Collectively, this design ensures high specificity, enabling the model to distinguish true lesion boundaries from background artifacts.

### 4.3. Ablation Study

We conduct a comprehensive ablation study on the ISIC 2018 dataset to validate both the incremental contributions of SGD-Net modules (Table 4) and the internal mechanisms of our SGDC module (Table 5). As shown in Table 4, the standard baseline (2) consisting of Res2Net + ViT exhibits a high HD95 of 32.63, indicating that semantic features alone lack sufficient spatial precision for accurate lesion delineation, notably, when the ViT component is removed (as seen in the Res2Net-only variant), the HD95 remains high at 30.54, further highlighting the baseline's struggle with boundary localization. In variant (3), we incorporate the Structural Guidance Encoder (SGE) into the baseline by adding the SGE-generated guidance feature to the baseline feature map via element-wise addition before feeding it to the decoder, which reduces the HD95 by 14.05 points to 18.58. This empirically confirms that explicit structural priors provide a crucial inductive bias that semantic features alone cannot capture.

To rigorously assess the effectiveness of the fusion strategy, we replace the simple addition used in variant (4) with Contmix (Lou and Yu, 2025), a recent dynamic convolution module that also leverages the SGE-generated guidance features. Despite its complexity, Contmix underperforms relative to the simple addition, yielding a higher HD95 of 24.21.

Table 6: Ablation study of different edge extraction operators on ISIC 2018. While Laplacian shows higher sensitivity, Sobel is selected for its structural stability and single-response characteristics.

| Operator | Type | Dice (%) | IoU (%) |
|---|---|---|---|
| No Edge Modulation | Baseline | 88.50 | 80.12 |
| Learnable Conv | Data-Driven | 89.85 | 82.40 |
| Scharr | Fixed (1st-order) | 90.44 | 83.57 |
| Laplacian | Fixed (2nd-order) | **92.11** | **85.26** |
| **Sobel (Ours)** | **Fixed (1st-order)** | 91.41 | 84.96 |

This outcome illustrates the "pooling trap": its adaptive pooling aggregates spatial context, inadvertently smoothing out high-frequency structural details provided by the SGE. In contrast, our SGDC module avoids pooling and employs a dual-branch design, enabling more precise integration of guidance features and leading to superior segmentation performance.

To further isolate the source of these gains, we analyze the internal components of SGDC in Table 5. First, we explicitly evaluated the model's robustness to weak or unavailable boundary annotations. Even when removing the explicit boundary guidance ($\lambda = 0$), the model maintains a competitive Dice of 90.37%. This validates that the fixed Sobel operator serves as an objective structural prior. Unlike learnable modules that are prone to collapse without supervision, this fixed prior prevents kernel degeneration, ensuring stability even when boundary labels are absent. However, the rise in HD95 (23.51) confirms that explicit supervision remains crucial for boundary precision. Second, regarding the guidance source, we implemented a Self-Guidance" variant where kernels are generated from the input feature $F_X$ itself, simulating standard implicit dynamic convolutions. This degradation leads to a sharp increase in HD95 (16.09 $\rightarrow$ 21.34), confirming that semantic features alone are insufficient for precise boundary delineation, and the performance gain stems specifically from the explicit structural prior. In the last, ablating the dual branches reveals a clear division of labor: removing the Local Branch causes a spike in HD95 (18.95), indicating that the static depthwise convolution acts as a deterministic "safety net" for high-frequency details, whereas removing the Dynamic Branch drops the Dice score (89.92), highlighting its role in capturing long-range semantic context. This confirms that SGDC succeeds through the complementary integration of stable local processing and adaptive dynamic modulation.

We analyzed different structural priors in Table 6. First, fixed operators consistently outperform learnable convolutions, indicating that learnable layers tend to overfit to dataset-specific semantics, whereas fixed operators provide the objective gradient signals necessary for stable modulation. Second, regarding the choice of fixed operator, While Laplacian achieves the highest Dice (92.11%), its second-order nature produces a zero-crossing double-response, creating ambiguity by highlighting boundary-adjacent regions. In contrast, Sobel provides a clean, single-response gradient. Although marginally lower in Dice, Sobel offers superior structural stability. Tests on CoNIC confirm that Laplacian degrades due to susceptibility to textural interference, whereas Sobel remains robust. Thus, we prioritize the geometric reliability of Sobel.

Table 7: Sensitivity analysis of the hyperparameter $\lambda$ on ISIC 2018 dataset. $\lambda = 3$ achieves the optimal trade-off.

| $\lambda$ Value | Dice (%) | IoU (%) |
|:---:|:---:|:---:|
| 0 | 90.37 | 83.25 |
| 1 | 90.00 | 82.74 |
| **3 (Ours)** | **91.41** | **84.96** |
| 5 | 90.23 | 82.91 |

Finally, we investigate the impact of the loss weight $\lambda$ in Eq. (5). As shown in Table 7, $\lambda = 3$ yields the optimal performance. This is theoretically grounded in our deep supervision design: since we employ three active segmentation heads (weights 1:1:1), a weight of $\lambda = 3$ for the single structural head ensures a macroscopic 1:1 magnitude balance between the total semantic objective and the structural objective. In contrast, that a lower weight ($\lambda = 1$) fails to provide sufficient guidance, resulting in high-noise gradients that degrade performance below the baseline ($\lambda = 0$). On the other hand, a higher weight ($\lambda = 5$) leads to over-regularization.

## 5. Conclusion

In this work, we have identified and addressed a critical limitation of spatially variant dynamic convolution methods, namely the reliance on adaptive average pooling for kernel generation leads to the degradation of fine-grained structural details. To overcome this bottleneck, our proposed SGD-Net incorporates the SGDC module, an average-pooling-free unit specifically designed to preserve high-frequency information during feature modulation.

Rather than compressing spatial contexts through aggregation, SGDC generates spatially precise kernels by leveraging structural priors provided by the auxiliary SGE branch, allowing dynamic modulation to adapt to local structure without sacrificing detail. Our experimental results demonstrate that this design effectively mitigates the over-smoothing problem inherent in pooling-based approaches, yielding notable improvements in boundary fidelity as measured by HD95.

Further internal analysis shows that the performance gains arise not merely from the inclusion of additional structural cues, but from the synergistic interaction between SGDC's deterministic local refinement and its adaptive dynamic modulation. We believe this strategy holds significant potential for broader vision tasks that require high fidelity in fine-grained structures, such as small object detection.

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
