# OpenReview forum: "SGDC: Structurally-Guided Dynamic Convolution for Medical Image Segmentation"
_MIDL.io/2026/Conference — MIDL 2026 Poster_

### Official Review · Reviewer_RGXe · 2025-12-29

**Confidence:** 4
**Preliminary Rating:** 4
**Final Rating:** 4

**Summary:**

This paper proposes SGD-Net, a medical image segmentation framework built around a Structure-Guided Dynamic Convolution module to address structural detail loss in pooling-based dynamic convolution. The core idea is to replace pooled semantic conditioning with explicitly supervised structural guidance, which is integrated through a dual-branch dynamic module combining adaptive modulation and local refinement. Experimental results and ablation studies show consistent improvements, particularly in boundary-sensitive metrics, supporting the effectiveness of pooling-free, structure-driven kernel generation.

**Strengths:**

- The paper clearly identifies a practical limitation of existing spatially variant dynamic convolution methods, namely the reliance on average pooling and semantically smooth conditioning signals that suppress high-frequency boundary information, which is well aligned with known challenges in medical image segmentation.
- The use of a fixed Sobel operator as a structural prior is a well-motivated design choice, avoiding unnecessary learning of basic gradient extraction and reducing the risk of semantic overfitting in boundary modeling.
- The architecture innovatively decouples the single-channel edge map (for explicit supervision) from the multi-channel structural guidance (for SGDC modules). This recognizes that representations for supervision (sharp contours) and guidance (multi-dimensional cues) are functionally different.

**Weaknesses:**

- The method relies on supervised boundary maps, but the paper does not sufficiently discuss how sensitive the approach is to boundary noise, or how it would generalize to datasets where reliable boundary annotations are unavailable or ambiguous.
- The baseline comparison focuses primarily on standard architectures and CTO. While reasonable, the evaluation would be more convincing if it included additional recent boundary-aware or kernel-adaptive segmentation methods that do not rely on dynamic convolution.
- Although class imbalance is briefly mentioned, skin lesion datasets are known to exhibit severe imbalance across lesion types and sizes. It remains unclear whether the proposed method performs consistently across different lesion characteristics or mainly benefits certain subclasses.

**Detailed Comments:**

- The authors choose fixed operators and state that learnable layers are prone to overfitting to specific semantic textures. However, the paper does not further analyze or empirically support this claim. A brief discussion or a small comparison with a learnable alternative in the SGE would help clarify whether the observed benefit arises from the fixed nature of the operator or from its specific mathematical formulation.
- The role of the Transformer encoder applied to shallow features is not deeply analyzed. An ablation removing or simplifying this component would help clarify its contribution relative to SGDC.
- While FLOPs and parameter counts are reported, a latency comparison on a standard GPU would be useful, as pixel-wise dynamic operations can introduce non-trivial runtime overhead.
- The claim that the local branch acts as a safety net against spatial instability is plausible, but the evidence is limited to performance metrics. A more direct or quantitative analysis of stability would strengthen this argument.
- The paper mentions potential applicability to small-object detection. Including even a limited experiment on a detection-oriented medical dataset would make this argument more concrete.

**Justification Of Final Rating:**

The rebuttal provides thoughtful clarifications and additional evidence that help better understand the design choices and empirical behavior of the proposed method. The authors have made an effort to address some of my questions, improving the clarity and presentation of the paper. However, the core methodological limitations identified in the original review (e.g., reliance on supervised boundary information, limited baseline coverage, and unclear generalization across imbalanced lesion characteristics) remain inherent to the current method design. I therefore maintain my original score.

**Justification Of The Preliminary Rating:**

The paper addresses a clear limitation of pooling-based dynamic convolution and proposes a structure-guided alternative that is carefully designed and empirically validated. However, the contribution largely builds on established operations and design patterns, and some aspects such as robustness to boundary supervision and broader generalization remain insufficiently explored.

**Questions To Address In The Rebuttal:**

1. How sensitive is SGDC to imperfect or noisy boundary annotations, and how would the method perform if boundary supervision were weak or unavailable?
2. Given the severe class imbalance in skin lesion datasets, do the authors observe consistent improvements across different lesion sizes or types, or are the gains primarily driven by specific subsets of the data?
3. Since SGDC operates on features at different spatial resolutions, could the authors clarify whether its benefits are consistent across scales? In particular, does SGDC have a stronger impact at coarser, semantic-rich levels or at finer, high-resolution levels, and how does the SGE guidance adapt to these varying resolutions?
4. Given that SGD-Net introduces additional parameters and dynamic operations beyond Baseline + SGE, could the authors clarify whether the remaining gains primarily stem from the proposed SGDC mechanism and its avoidance of the pooling trap, or from increased model capacity and training complexity? A parameter- or complexity-matched comparison would help better isolate the specific contribution of the dynamic kernels.

---

> ### Author Response · Authors · 2026-01-25
> **Responses to Questions 1–4**
>
> **Q1**
> Reply: We thank the reviewer for this important question. We address two aspects separately: (1) robustness to noisy annotations, and (2) performance when boundary supervision is weak or unavailable.
>
> First, SGD-Net is robust to imperfect boundary labels. In SGE, the sobel operator extracts structural cues from shallow features and semantic cues from deep features. The deep semantic features then modulate the shallow structural features, producing guidance that is less affected by noisy boundary labels and artifacts. This is included in Section 4.2 of the revised manuscript.
>
> SGD-Net remains effective even when explicit boundary supervision is removed. In the ablation with λ_edge = 0, as reported in Table 5 of the revised manuscript, the model still achieves 90.37% Dice, showing that the semantic supervision in SGE can still produce useful structural guidance for SGDC.
>
> | Method                      | Dice ↑ | IoU ↑ | HD95 ↓ |
> |----------------------------|--------|-------|--------|
> | SGD-Net (Ours)             | 91.41  | 84.96 | 16.09  |
> | w/o Boundary Supervision   | 90.37  | 83.25 | 23.51  |
>
> **Q2**
> Reply: We thank the reviewer for highlighting the critical issue of class imbalance. Regarding the consistency of improvements, our observations indicate that the improvements are not solely driven by a single subset of cases. On ISIC 2018, the performance gains mainly arise from better boundary structure preservation, which is consistent with the dataset’s dominant challenge of boundary ambiguity. On CoNIC, we observe that the improvement is partly due to boundary structure preservation and partly due to a reduction in false positives in complex background regions. This is consistent with the role of SGE’s semantic modulation. This is included in Section 4.2 of the revised manuscript.
>
>
> **Q3**
> Reply: We thank the reviewer for this question. At finer, high-resolution scales, SGDC mainly focuses on boundary structure preservation. At coarser, semantic-rich scales, SGDC functions more as a semantic-boundary preservation mechanism. It prevents over-smoothing of deep semantic regions, thereby maintaining topological and semantic integrity. Overall, we observe that the fine-scale SGDC has a slightly stronger impact on boundary refinement, while the coarse-scale SGDC helps preserve semantic structure. The SGE guidance adapts across resolutions through bidirectional interpolation and channel-wise 1×1 convolution, enabling consistent and resolution-aware structural cues for SGDC.
>
> **Q4**
> Reply: We appreciate the reviewer’s careful review regarding the source of our improvements. The remaining performance gains primarily stem from the proposed SGDC mechanism and its ability to avoid the pooling trap, rather than from increased model capacity or training complexity. This is supported by several observations. First, in terms of parameter efficiency, SGD-Net contains approximately 75M parameters, which is about 34% fewer than TransUNet(114M parameters). Despite its smaller size, SGD-Net outperforms TransUNet in both Dice and IoU on the ISIC 2018 dataset. Second, increasing model capacity mainly improves semantic recognition (reflected in Dice) but cannot correct the boundary blurring induced by average pooling operations. In contrast, SGD-Net achieves a substantial reduction in HD95 (32.63 → 16.09). Third, the performance gain is accompanied by improved specificity (reduced false positives), suggesting that the improvement is related to structural modulation and noise rejection rather than merely increased capacity.

---

> ### Author Response · Authors · 2026-01-25
> **Response to W1-W3**
>
> **W1**
> Reply: This concern is addressed in Q1. Please see the corresponding responses for the detailed explanation.
>
> **W2**
> Reply: We thank the reviewer for the suggestion. In our manuscript we have compared with standard segmentation architectures as well as CTO (a 2025 Medical Image Analysis paper), which represents one of the current state-of-the-art methods with explicit boundary-aware design. Our choice of baselines reflects both common practice in skin lesion segmentation and representative approaches to structural supervision. The superior performance of our method over these baselines, particularly on boundary-sensitive metrics (e.g., HD95, specificity), supports the effectiveness of our structural guidance design.
> We will consider extending the comparison to additional boundary-aware architectures and other kernel-adaptive segmentation in our future works.
>
> **W3**
> Reply: This concern is addressed in Q2. Please see the corresponding responses for the detailed explanation.

---

> ### Author Response · Authors · 2026-01-25
> **Response to detailed comments**
>
> **D1**
> Reply: We thank the reviewer for this insightful suggestion. We have added a direct ablation of different structural priors in Table 6 of the revised manuscript. The results show that fixed operators consistently outperform learnable convolutions, indicating that learnable layers tend to overfit to dataset-specific semantics, while fixed operators provide more stable and objective gradient signals for modulation. Detailed results are presented in the table below.
>
> | Method          | Operator Type        | Dice (%) / IoU (%) |
> |-----------------|----------------------|--------------------|
> | Learnable Conv  | Data-Driven          | 89.85 / 82.40      |
> | **Sobel (Ours)**| Fixed (1st-order)    | **91.41 / 84.96**  |
>
> **D2**
> Reply: We thank the reviewer for pointing this out. We agree that it is important to isolate the contribution of the Transformer encoder on shallow features. To clarify this, we conducted an ablation study by removing the Transformer encoder while keeping all other components unchanged. The results are shown in Table 4 of revised manuscript and below:
> | Method                             | Dice (%) ↑ | IoU (%) ↑ | HD95 ↓ |
> |------------------------------------|------------|-----------|--------|
> | (1) Res2Net                        | 89.13      | 82.42     | 30.54  |
> | (2) Baseline (Res2Net + ViT)       | 89.57      | 82.91     | 32.63  |
> Incorporating the Transformer encoder with shallow features results in only modest gains in Dice (+0.44) and IoU (+0.49), while HD95 deteriorates (+2.09). This suggests that the Transformer encoder alone offers limited structural enhancement and may even impair boundary localization. Consequently, the performance improvement of our full model is primarily attributed to the SGDC module, which explicitly integrates structural guidance into dynamic convolution.
>
> **D3**
> Reply: We thank the reviewer for the suggestion. We agree that latency is an important metric for practical deployment. However, the main contribution of this paper is methodological. In our current implementation, the Transformer auxiliary branch and the dual-attention module are only used during training and are not involved in inference. As our code is a research prototype without inference-level optimization, we report FLOPs and parameter counts as complexity metrics, and leave latency optimization (e.g., kernel fusion or TensorRT) to future work.
>
> **D4**
> Reply: We thank the reviewer for the suggestion. Dedicated stability metrics are not included in this revision and will be investigated in our future work. Nevertheless, our ablation shows that removing the local branch increases HD95 from 16.09 to 18.95, while Dice/IoU only slightly decrease, indicating the local branch helps reduce boundary deviations and improve spatial consistency. This supports its role as a safety net against spatial instability.
>
> **D5**
> Reply: We thank the reviewer for the suggestion. Extending SGDC to small-object detection is an interesting direction, but it is beyond the scope of the current segmentation-focused work. We plan to explore this extension in future work to validate the applicability of our approach.

---

### Official Review · Reviewer_9LLd · 2026-01-06

**Confidence:** 5
**Preliminary Rating:** 2
**Final Rating:** 4

**Summary:**

This paper proposes a structure-guided, pooling-free spatially-variant dynamic convolution for medical image segmentation, motivated by the observation that average-pooling-based kernel generation can discard high-frequency details and lead to over-smoothed boundaries. Experiments on ISIC 2016→PH2, ISIC 2018 (5-fold CV), and CoNIC show improved Dice/IoU and notably improved boundary fidelity (HD95), supported by ablations that isolate the benefit of explicit structure guidance and the dual-branch SGDC design.

**Strengths:**

1.	Clear motivation. The paper articulates a concrete failure mode (structural detail loss from pooling-conditioned dynamic kernels) and proposes an architectural change that directly targets it: replacing context aggregation with pixel-wise structural guidance for kernel/gate generation.

2.	SGDC’s dual-path refinement (dynamic unfold–weight–fold branch + local depthwise conv branch) is well motivated as “complementary processing,” and the formulation is easy to follow.

3.	Experimental on several datasets demonstrate the effectiveness of proposed method.

**Weaknesses:**

1	Comparison set may be incomplete for the claimed “state-of-the-art”. For example, on ISIC 2018, the paper does not compare with UNet v2 and VM-UNetV2.

2	Important reproducibility items are missing or not explicit in the visible text: how boundary ground truth E_gtis derived (thickness, morphological operator, resolution), augmentation pipeline, decoder design, kernel size / neighborhood size used by unfold–fold, and memory/runtime overhead. What are the GPUs used for the experiments?

3	The argument that learnable edge extraction may “overfit” is plausible, a direct ablation (Sobel vs. learnable conv / Laplacian / Scharr / no edge modulation) would better support the design choice.

4	For ISIC 2018 5-fold CV, can you report mean ± std (or confidence intervals) across folds/runs, especially since improvements are modest in Dice/IoU?

**Detailed Comments:**

See weakness.

**Justification Of Final Rating:**

The authors have addressed most of my concerns. For example, adding UNet v2 and VM-UNetV2 comparisons, ablation (Sobel vs. learnable conv / Laplacian / Scharr / no edge modulation), and important reproducibility details. Therefore, I would like change my final rating from 2 to 4.

**Justification Of The Preliminary Rating:**

This paper proposes a structure-guided, pooling-free spatially-variant dynamic convolution for medical image segmentation, motivated by the observation that average-pooling-based kernel generation can discard high-frequency details and lead to over-smoothed boundaries. Experiments on ISIC 2016→PH2, ISIC 2018 (5-fold CV), and CoNIC show improved Dice/IoU and notably improved boundary fidelity (HD95), supported by ablations that isolate the benefit of explicit structure guidance and the dual-branch SGDC design. However, there are still some concerns listed in weakness. Therefore, I suggest weak reject.

**Questions To Address In The Rebuttal:**

See weakness.

---

> ### Author Response · Authors · 2026-01-25
> **Response to W1**
>
> We sincerely thank the reviewer for pointing out these important state-of-the-art methods. We agree that comparing with broader baselines like UNet v2 and VM-UNetV2 strengthens the evaluation of our proposed method.
>
> We evaluate all three methods under two settings for completeness and fair comparison. Setting 1 (No TTA, No post-processing) and Setting 2 (With TTA + post-processing), setting 2 follows the evaluation protocol used in the paper, where all methods apply the same TTA and post-processing. Our TTA aggregates predictions from 8 views (horizontal/vertical flips and 90°/180°/270° rotations). Post-processing uses RSA and ABL.  Results are shown in the table below. We have incorporated these comprehensive comparisons into Table 2 of the revised manuscript.
>
> | Method | Setting 1 |  | Setting 2 |  |
> |--------|-----------------------------|--|----------------------|--|
> |        | Dice (%) | IoU (%) | Dice (%) | IoU (%) |
> | UNet v2 | 89.33 | 80.66 | 90.47 | 82.63 |
> | VM-UNetV2 | 89.68 | 81.10 | 90.79 | 83.31 |
> | **SGD-Net (Ours)** | **90.22 ± 0.49** | **82.24 ± 0.82** | **91.41 ± 0.24** | **84.96 ± 0.13** |

---

> ### Author Response · Authors · 2026-01-25
> **Response to W2**
>
> Thank you for your comments.
> To facilitate reproducibility, we clarify key settings.
> The boundary ground truth E_gt is generated by the morphological gradient (dilation minus erosion) using a 3×3 structuring element, yielding a fixed 3-pixel thickness. The SGDC module uses a 7×7 neighborhood for the dynamic unfold–fold operation. Data augmentation (Albumentations) includes resizing to 256×256, random horizontal/vertical flips (p = 0.5), and random rotation within ±20° (p = 0.7). All experiments were conducted on a single NVIDIA L4 GPU. These details are also provided in the revised manuscript (Section 4.1 and Section 3.1).

---

> ### Author Response · Authors · 2026-01-25
> **Response to W3**
>
> We thank the reviewer for this insightful suggestion. We have added a direct ablation of different structural priors in Table 6 and section4.3 of the revised manuscript. The results show that fixed operators consistently outperform learnable convolutions, indicating that learnable layers tend to overfit to dataset-specific semantics, while fixed operators provide more stable and objective gradient signals for modulation.
>
> Among fixed operators, Laplacian achieves the highest Dice (92.11%) but its second-order nature produces a zero-crossing double-response, which can highlight boundary-adjacent regions and introduce ambiguity. In contrast, Sobel generates a clean single-response gradient, and although it is marginally lower in Dice, it offers superior structural stability. This is further supported by results on CoNIC, where Laplacian degrades compared to Sobel (Dice ↓ 0.7%, IoU ↓ 0.6%). Therefore, we prioritize the geometric reliability of Sobel in our final design.
>
> | Method               | Operator Type        | Dice (%) / IoU (%) |
> |----------------------|----------------------|--------------------|
> | No Edge Modulation   | Baseline             | 88.50 / 80.12      |
> | Learnable Conv       | Data-Driven          | 89.85 / 82.40      |
> | Scharr               | Fixed (1st-order)    | 90.44 / 83.57      |
> | Laplacian            | Fixed (2nd-order)    | 92.11 / 85.26      |
> | **Sobel (Ours)**     | Fixed (1st-order)    | **91.41 / 84.96**  |

---

> ### Author Response · Authors · 2026-01-25
> **Response to W4**
>
> The results reported in Table 2 of the revised manuscript are obtained from 5-fold cross-validation on ISIC 2018, and we report mean ± standard deviation across the 5 folds. SGD-Net achieves 90.22 ± 0.49% Dice and 82.24 ± 0.82% IoU under Setting 1, and 91.41 ± 0.24% Dice and 84.96 ± 0.13% IoU under Setting 2.

---

### Official Review · Reviewer_W3HR · 2026-01-08

**Confidence:** 4
**Preliminary Rating:** 3
**Final Rating:** 4

**Summary:**

This paper proposes SGDC (Structurally-Guided Dynamic Convolution) for medical image segmentation. They want to tackle the issues brought by the adaptive average pooling, like oversmoothed predictions. They propose SGDC to guide the pooling mechanism, and thus improve the final performance in object detection tasks. The results looks good, but I still have some concerns.

**Strengths:**

1. The starting point about the concern related to average pooling makes sense, which leads to an interesting work.
2. The SGDC module design logically makes sense. The guidance-driven kernel + dual-branch computation (dynamic + depthwise local refinement) is a reasonable way to combine adaptivity with stability.
3. The evaluations are relatively comprehensive, across multiple datasets and different organ/tissues. The baselines are relatively new, containing works done in 2024 and 2025.

**Weaknesses:**

1. I have a concern on the added computational complexity in order to improve the issue brought by the pooling operation. I saw Transformer encoders and Dual Attentions. Will these lead to very heavy overhead? I’m assuming the “light weight” from average pooling could be one of the major reasons why major models using average pooling rather than more complex models
2. If possible, I would like to see a brief ablation regarding how the Lambda=3 was selected. Since there are three loss terms, I’m curious to see how the balances were made.
3. “We conduct a comprehensive ablation study on the ISIC 2018 dataset” Do the observations also hold for other datasets? No need to list all numbers (but it would be better to have them), just want to ask the authors to have a brief discussion on it. This is because I saw your two ablations are both done in the ISIC 2018 dataset, but not the other ones.

**Detailed Comments:**

N/A

**Justification Of Final Rating:**

I have updated my rating since the authors responded me well on the concerns I had.
I have updated my rating since the authors responded me well on the concerns I had.
I have updated my rating since the authors responded me well on the concerns I had.

**Justification Of The Preliminary Rating:**

Strength:
1. The starting point about the concern related to average pooling makes sense, which leads to an interesting work.
2. The SGDC module design logically makes sense. The guidance-driven kernel + dual-branch computation (dynamic + depthwise local refinement) is a reasonable way to combine adaptivity with stability.
3. The evaluations are relatively comprehensive, across multiple datasets and different organ/tissues. The baselines are relatively new, containing works done in 2024 and 2025.
Weakness:
1. I have a concern on the added computational complexity in order to improve the issue brought by the pooling operation. I saw Transformer encoders and Dual Attentions. Will these lead to very heavy overhead? I’m assuming the “light weight” from average pooling could be one of the major reasons why major models using average pooling rather than more complex models
2. If possible, I would like to see a brief ablation regarding how the Lambda=3 was selected. Since there are three loss terms, I’m curious to see how the balances were made.
3. “We conduct a comprehensive ablation study on the ISIC 2018 dataset” Do the observations also hold for other datasets? No need to list all numbers (but it would be better to have them), just want to ask the authors to have a brief discussion on it. This is because I saw your two ablations are both done in the ISIC 2018 dataset, but not the other ones.

In all, I gave out my rating.

**Questions To Address In The Rebuttal:**

Please see the weakness section.

---

> ### Author Response · Authors · 2026-01-25
> **Response to W1**
>
> We appreciate the reviewer's attention to computational complexity. \
> Fisrt, we would like to clarify a potential misunderstanding regarding the Transformer and Dual Attention implementation: specifically, the Transformer Encoder employs an efficient sparse sampling strategy with multi-scale strides (16,16), (8,8), (4,4), (2,2), which dramatically reduces sequence length and computational cost. It contributes only **8.07 GFLOPs** to the total model complexity. Similarly, the Dual Attention module is applied only once at the deepest encoder layer where feature resolution is lowest ($8\times8$), adding a mere **2.01 GFLOPs**. These implementations are adapted from the CTO [1] framework we cited. We acknowledge that these specifics were omitted initially and have been added in Section 3.1 of the revised manuscript for clarity.
>
> Second, regarding the role of average pooling, we should mention that its widespread adoption in dynamic convolution architectures is largely driven by the need for effective feature aggregation and receptive field expansion, rather than solely for computational efficiency. In established dynamic convolution paradigms, such as CondConv[2] and DynamicConv[3], pooling operations are employed to aggregate semantic context to condition the generation of adaptive kernels. While the low computational cost is a beneficial side effect, the primary objective is to capture semantic representations. However, this aggregation comes at the cost of spatial precision, as it inevitably collapses high-frequency structural details. Our SGDC mechanism is specifically designed to resolve this trade-off: by replacing the semantic compression of pooling with explicit structural guidance, we achieve superior boundary fidelity (reducing HD95 from 32.63 to 16.09). This demonstrates that replacing average pooling is not a complication, but a necessity for preventing the over-smoothing of fine-grained medical structures.
>
> [1] Yi Lin, Dong Zhang, Xiao Fang, Yufan Chen, Kwang-Ting Cheng, and Hao Chen. Rethinking boundary detection in deep learning-based medical image segmentation. Medical Image Analysis, page 103615, 2025.
>
> [2] Brandon Yang, Gabriel Bender, Quoc V Le, and Jiquan Ngiam. Condconv: Conditionally parameterized convolutions for efficient inference. Advances in neural information processing systems, 32, 2019.
>
>
> [3] Yinpeng Chen, Xiyang Dai, Mengchen Liu, Dongdong Chen, Lu Yuan, and Zicheng Liu. Dynamic convolution: Attention over convolution kernels. In Proceedings of the IEEE/CVF conference on computer vision and pattern recognition, pages 11030–11039, 2020.

---

> ### Author Response · Authors · 2026-01-25
> **Response to W2**
>
> We appreciate the reviewer’s insightful comments.
> First, we would like to respectfully clarify the composition of our objective function.
> As explicitly defined in Eq. (5), the total loss consists of four distinct terms, rather than three.
> L_total = sum_{i=1}^3 L_seg^(i) + L_edge
> Specifically, the first three terms correspond to the segmentation losses applied via deep supervision at three different decoder scales, while the fourth term is an edge loss derived from the SGE branch.
>
> The choice of λ = 3 is motivated by both architectural necessity and empirical stability. Our network employs three deep supervision segmentation heads with equal weights (1:1:1), resulting in a total semantic supervision magnitude of 3. Setting λ = 3 therefore establishes a balanced 1:1 ratio between semantic and structural objectives. More importantly, the edge prediction from the SGE branch directly serves as guidance for subsequent SGDC modules. Insufficient supervision on this branch leads to unstable training due to low-quality structural guidance.
>
> We have added a brief ablation of λ in the revised manuscript  Table.7 (Section 4.3). Detailed results are presented in the table below.
>
> | λ Value | Dice (%) | IoU (%) |
> |--------|---------|---------|
> | 0      | 90.37   | 83.25   |
> | 1      | 90.00   | 82.74   |
> | **3 (Ours)** | **91.41** | **84.96** |
> | 5      | 90.23   | 82.91   |

---

> ### Author Response · Authors · 2026-01-25
> **Response to W3**
>
> Thank you for your insightful question.
>
> We selected ISIC 2018 as the primary testbed for the ablation study because it provides the most direct environment to isolate and validate our core hypothesis: resolving the "over-smoothing" of boundaries caused by pooling. In ISIC, the dominant challenge is boundary ambiguity, which allows to clearly show performance gains due to the structure-preserving capabilities of SGE and SGDC.
>
> We confirm that the efficacy of our modules holds consistently on other datasets like CoNIC, where our method achieves state-of-the-art performance. However, the performance gains on CoNIC are multifaceted and stem from a combination of factors. Beyond the direct structure-preserving capability of the core modules, a portion of the improvement is attributed to the synergistic effect between these modules and the decoder. The decoder design, originally intended to enhance noise resistance, interacts with the core modules to produce a strong background suppression effect. This makes CoNIC less ideal for a “clean” ablation study focused solely on the core modules.

---

### Author Rebuttal · Authors · 2026-01-25

**Rebuttal:**

We are grateful for the reviewer's constructive comments. We have addressed the reviewers' concerns in each comment section of respective reviewers, with changes highlighted in colored text in our revised manuscript (Red for Reviewer W3HR, Green for Reviewer 9LLd, and Lavender for Reviewer RGXe). The light-yellow highlights correspond only to minor grammatical edits.

We are attaching the revised manuscript as the supporting material here.

**Supporting Material:**

/attachment/2a417a07b921c3f3264e0e18a710cdb85f74b959.pdf

---

### Author Response · Authors · 2026-02-02

Dear [Program Chairs / AC],
First, I would like to sincerely thank you for providing the platform to engage in the discussion phase for our submission (#79). I also appreciate the efforts of the two reviewers who have already updated their feedback.
We would be grateful if the remaining reviewer 9LLd could kindly provide any further comments or clarifications regarding our rebuttal, if possible. Their comment would be very helpful for the ongoing discussion and for ensuring a thorough and fair evaluation of our paper.
Thank you for your time and support.

---

### Meta-Review · Area_Chair_RMYC · 2026-02-08

**Recommendation:** Accept (Poster)
**Confidence:** 5

**Metareview:**

This is an interesting work which proposed a new dynamic convolution method for medical image segmentation. The main contribution of this work is the specific design of the structure-guided convolution for the segmentation task and the evaluation can also support the claims from the authors. After the rebuttal phase, all reviewers agree that this manuscript worth the presentation in MIDL. Therefore, my recommendation is accept.

---

### Decision · Program_Chairs · 2026-02-13

Accept (Poster)